# Long-term outcome of prolonged critical illness: A multicentered study in North Brisbane, Australia

Kevin B. Laupland[1,2]*, Mahesh Ramanan[3,4], Kiran Shekar[3,5], Felicity Edwards[2], Pierre Clement[1], Alexis Tabah[3,6]

1 Department of Intensive Care Services, Royal Brisbane and Women's Hospital, Brisbane, Queensland, Australia, 2 Queensland University of Technology (QUT), Brisbane, Queensland, Australia, 3 Faculty of Medicine, University of Queensland, Brisbane, Queensland, Australia, 4 Intensive Care Unit, Caboolture Hospital, Caboolture, Queensland, Australia, 5 Intensive Care Unit, The Prince Charles Hospital, Brisbane, Queensland, Australia, 6 Intensive Care Unit, Redcliffe Hospital, Redcliffe, Queensland, Australia

* Kevin.laupland@qut.edu.au

## Abstract

### Background

Although critical illness is usually of high acuity and short duration, some patients require prolonged management in intensive care units (ICU) and suffer long-term morbidity and mortality.

### Objective

To describe the long-term survival and examine determinants of death among patients with prolonged ICU admission.

### Methods

A retrospective cohort of adult Queensland residents admitted to ICUs for 14 days or longer in North Brisbane, Australia was assembled. Comorbid illnesses were classified using the Charlson definitions and all cause case fatality established using statewide vital statistics.

### Results

During the study a total of 28,742 adult Queensland residents had first admissions to participating ICUs of which 1,157 (4.0%) had prolonged admissions for two weeks or longer. Patients with prolonged admissions included 645 (55.8%), 243 (21.0%), and 269 (23.3%) with ICU lengths of stay lasting 14–20, 21–27, and ≥28 days, respectively. Although the severity of illness at admission did not vary, pre-existing comorbid illnesses including myocardial infarction, congestive heart failure, kidney disease, and peptic ulcer disease were more frequent whereas cancer, cerebrovascular accidents, and plegia were less frequently observed among patients with increasing ICU lengths of stay lasting 14–20, 21–27, and ≥28 days. The ICU, hospital, 90-day, and one-year all cause case-fatality rates were 12.7%, 18.5%, 20.2%, and 24.9%, respectively, and were not different according to duration

**Data Availability Statement:** Data cannot be shared publicly because of institutional ethics, privacy, and confidentiality regulations. Data release for the purposes of research under Section

280 of the Public Health Act 2005 requires application to the Director General (PHA@health. qld.gov.au).

**Funding:** This study was supported by a grant from the Queensland University of Technology (QUT) awarded to KL https://www.qut.edu.au. The funders had no role in study design, data collection and analysis, decision to publish, or preparation of the manuscript.

**Competing interests:** The authors have declared that no competing interests exist.

of ICU stay. The median duration of observation was 1,037 (interquartile range, 214–1888) days. Although comorbidity, age, and admitting diagnosis were significant, neither ICU duration of stay nor severity of illness at admission were associated with overall survival outcome in a multivariable Cox regression model.

## Conclusions

Most patients with prolonged stays in our ICUs are alive at one year post-admission. Older age and previous comorbidities, but not severity of illness or duration of ICU stay, are associated with adverse long-term mortality outcome.

## Introduction

Critical illness is characterized by both high severity and acuity of disease with the majority of deaths occurring within the first 2–3 days of admission to intensive care units (ICU) [1]. However, some patients require admission to ICU for prolonged periods of 2 weeks duration or longer with extreme cases resulting in lengths of stay lasting 3 or more months [2, 3]. Studies conducted over the past two decades report that while only <10% of patients admitted to ICU require prolonged stays, these patients suffer higher case-fatality and greater functional impairment as compared to patients with shorter stays [1, 4, 5]. Although admission diagnosis and severity of illness are important predictors of short term mortality, demographic and comorbid medical conditions are more important predictors of prolonged ICU stay and subsequent survival outcome [3, 6].

It is important to establish the occurrence and determinants of patients requiring prolonged ICU legth of stays to inform decision making and prognostication. While there is an increasing body of literature addressing this topic area, studies have been limited by a number of methodological issues including small sample size, study of specific conditions, sub-populations, or age groups, or assessment of short-term or in-hospital death only [7–11]. In addition, while indicies inclusive of a range of underlying illnesses have been widely used for determination of comorbid illnesses in medical research, outcome studies in ICUs have largely been limited to evaluation of a small number of selected chronic diseases routinely measured in severity of illness measures [12–15].

The objective of this study was to examine prolonged admission to ICUs in a large population of critically ill patients with broad case-mix and examine the subsequent long-term survival related to their acute and chronic illness characteristics.

## Materials and methods

### Study design

Retrospective multi-centred cohort with statewide database linkages.

### Study sites and subjects

Study sites included all four publicly funded, closed-model, medical-surgical ICUs within the Metro North Hospital and Health Service area in Queensland, Australia [16]. These ICUs are staffed by specialists certified by the College of Intensive Care Medicine of Australia and New Zealand. Although a small number of children may be admitted to these ICUs they are primarily focussed on management of adults. The Royal Brisbane and Women's Hospital (RBWH) is

a ≈1000-bed urban academic institution (33 ICU beds) which serves as a major neurosurgical, trauma, and burns referral centre for the state. The Prince Charles Hospital (TPCH) is a 630-bed urban teaching hospital (27 ICU beds) that is a major cardiac, respiratory, and cardiothoracic surgical referral hospital and is the state heart and lung transplant centre. Redcliffe Hospital and Caboolture Hospital are 270-bed (10 ICU beds) and 265-bed (6 ICU beds) regional institutions serving north Brisbane.

Patients with prolonged (2 weeks and longer) admissions were included. Admissions were limited to first ICU admissions following study inception and to Queensland residents aged 18 years and older. Inception dates for ICU admissions varied due to the availability of electronic data and were January 2012 for both RBWH and TPCH, March 2016 for Redcliffe Hospital, and April 2016 for Caboolture Hospital with inclusion through to and including December 31, 2019. This study was approved by the RBWH Human Research Ethics Committee (LNR/2019/QRBW/58463) with an individual waiver of consent granted. While unique patient identifiers were required for data linkage purposes, all files were fully anonymized before access for analysis purposes.

## Study protocol

Admissions were identified and base information obtained independently from the clinical information systems available at each of the four sites. The eCritical MetaVision™ (iMDsoft, Boston, MA, USA) system was used for patients admitted to ICUs at Caboolture Hospital, Redcliffe Hospital, and RBWH, with data prior to 2014 from RBWH obtained using IntelliVue Clinical Information Portfolio (ICIP, Philips Healthcare, Amsterdam, NL). The Core Outcome Measurement and Evaluation Tool (COMET) application was used for patients admitted to TPCH [17]. Standard Australia and New Zealand Intensive Care Society (ANZICS) Centre for Outcome and Resource Evaluation Adult Patient Database definitions and dictionary was used for all sites [18].

Further diagnostic and death outcome was obtained through linkages to the state-wide Queensland Hospital Admitted Patient Data Collection (QHAPDC) and Registry of General Deaths [19]. Data from the QHAPDC included diagnostic codes (ICD-10AM) and dispositions associated with all admissions to private and public hospitals in Queensland from 2012 to end of inception. Diagnostic categories were based on the major diagnostic group classifications with pooling of categories 1, 19, and 20 into "neurologic"; 2 and 3 into "head and neck"; 6 and 7 into "gastrointestinal", 8 and 9 as "soft tissue"; 11 to 14 as "genitourinary"; and 16 and 17 as "blood/neoplastic" [20]. Comorbid illnesses were established and weighted according to the definitions of Charlson *et al* by applying the validated algorithms developed by Sundararajan *et al* [12, 21]. Only ICD-10AM codes from hospital separations including or preceeding the index ICU admission were used. Vital status was established for all study participants as of March 31, 2020 using the Registry of General Deaths.

## Data management and analysis

Data were managed and analysed using Access 2016 (Microsoft, Redmond, USA) and Stata 16 (StataCorp, College Station, USA), respectively. Days of stay were counted as calendar days or part thereof and were grouped according to *a priori* specified categories of 14–20, 21–27, and ≥28 days. Prior to analysis continuous variables were plotted using histograms to assess their underlying distribution. Skewed variables were reported as medians with interquartile ranges (IQR). The non-parametric test for trend (nptrend) test was used to compare ordered groups by ICU admission duration gouping. A Kaplan-Meier plot and logrank test was used to display and test equality of the survivor functions, respectively. A multivariable Cox regression model

was developed in order to examine factors associated with survival following ICU admission. Variables included in the initial model were grouped ICU length of stay, Charlson score, age, sex, major diagnostic category, and APACHE III score. Variables were then eliminated in a stepwise fashion in order to develop the most parsimonius model. The final model was assessed for fulfillment of the assumptions of constant hazards using analysis of scaled Schoenfeld residuals. In all comparisons a p-value of less than 0.05 was deemed to represent significance.

## Results

During the study a total of 28,742 adult Queensland residents had first admissions to participating ICUs. Among this cohort, 17,522 (60.9%) were male, the median age was 62.0 (IQR, 48.2–72.0) years, and the median APACHE III score was 133 (IQR, 119–149). The median length of stay in ICU was 2 (IQR, 2–4) calendar days. Within this overall cohort, 1,157 (4.0%) had prolonged admissions for more two weeks or longer and comprised the study cohort for all other analyses. Patients with prolonged admissions included 645 (55.8%), 243 (21.0%), and 269 (23.3%) with ICU lengths of stay lasting 14–20, 21–27, and ≥28 days, respectively. Among those staying ≥28 days, the median stay was 37 (IQR, 31–46) with 31 and 11 patients staying ≥60 and ≥90 days, respectively.

A number of different baseline characteristics were observed according to subsequent prolonged admission category as shown in Table 1. Neither the severity of illness at admission as measured by the APACHE III score nor age were associated with ICU length of stay categories. Although none of the APACHE III chronic disease categories were associated, the presence of several comorbid illnesses as defined by the Charlson classification were different among those with different lengths of stay in ICU. Most notably, myocardial infarction, congestive heart failure, renal disease, and peptic ulcer disease increased whereas cancer, cerebrovascular accident, and plegia decreased proportionally across the ICU lengths of stay categories (Table 1). There was a significant difference in length of ICU stay based on main diagnostic groups. When categorized by stays of 14–20, 21–27, and ≥28 days, the proportion of neurological diagnoses decreased and respiratory and burns diagnoses increased (p≤0.001 for each).

Among the patients who stayed for 14–20, 21–27, and ≥28 days in ICU, the ICU case-fatality rates were 12.6% (81/645), 14.0% (34/243), and 11.9% (32/269); p = 0.896; and and in-hospital case-fatality rates were 18.1% (117/645), 21.4% (52/243), and 16.7% (45/269); p = 0.822, respectively. Among the 943 survivors to hospital separation, the majority (488; 51.7%) were discharged home and there was significant differences (overall p<0.001) in the disposition by duration of ICU stay as shown in Table 2.

All cause case-fatality was 20.2% (234/1157) at 90-days post-ICU admission, and was not significantly different (p = 0.333) among those who stayed for 14–20, 21–27, and ≥28 days in ICU at 20.6% (133/645), 22.6% (55/243), and 17.1% (46/269), respectively. Among the cohort admitted to ICU prior to March 31, 2019 for which at least one full year of follow-up information was available, the overall one-year all cause case-fatality was 24.9% (264/1,061) and was not different based on duration of ICU stay (144/596, 24.2%; 61/218, 28.0%; and 59/247, 23.9% for stays lasting 14–20, 21–27, and ≥28 days; p = 0.875, respectively). Within the overall cohort of 1,157 patients with at least 90 days of follow-up, the median duration of observation was 1,037 (IQR, 214–1888) days. The overall survival did not significantly (p = 0.883) vary by duration of ICU stay as shown in Fig 1.

A multivariable Cox regression model was developed for mortality outcome among patients who required prolonged admission to ICU and the results are displayed in Table 3. Neither the

**Table 1. Admission characteristics of patients with prolonged admission to ICU.**

| Variable | ICU stay 14–20 days (n = 645) | ICU stay 21–27 days (n = 243) | ICU stay ≥28 days (n = 269) | p-value |
|---|---|---|---|---|
| Median years of age (interquartile range, IQR) | 56.4 (43.9–68.0) | 61.1 (43.4–70.1) | 58.3 (40.6–66.6) | 0.934 |
| Male | 395 (61.2%) | 155 (63.8%) | 187 (69.5%) | 0.020 |
| Median APACHE III (IQR) | 146 (125–165) | 147 (123–169) | 149 (128–166) | 0.702 |
| APACHE III Comorbidities | | | | |
| AIDS | 1 (0.2%) | 2 (0.8%) | 0 | 0.986 |
| Hepatic Failure | 5 (0.8%) | 0 | 2 (0.7%) | 0.74 |
| Lymphoma | 5 (0.8%) | 0 | 1 (0.4%) | 0.311 |
| Metastatic cancer | 7 (1.1%) | 2 (0.8%) | 0 | 0.099 |
| Leukaemia/Myeloma | 10 (1.6%) | 3 (1.2%) | 2 (0.7%) | 0.326 |
| Immunosuppressed | 0 | 1 (0.4%) | 0 | 0.177 |
| Cirrhosis | 13 (2.0%) | 2 (0.8%) | 3 (1.1%) | 0.234 |
| Median Charlson Co-morbidity Index (IQR) | 2 (1–3) | 2 (1–3) | 2 (0–4) | 0.215 |
| Charlson variables | | | | |
| MI | 79 (12.3%) | 43 (17.7%) | 51 (19.0%) | 0.005 |
| CHF | 114 (17.7%) | 52 (21.4%) | 66 (24.5%) | 0.015 |
| PVD | 66 (10.2%) | 39 (16.1%) | 33 (12.3%) | 0.194 |
| CVA | 168 (26.1%) | 49 (20.2%) | 45 (16.7%) | 0.001 |
| Plegia | 98 (15.2%) | 37 (15.2%) | 21 (7.8%) | 0.006 |
| Pulmonary | 119 (18.5%) | 59 (24.3%) | 57 (21.2%) | 0.205 |
| Any DM | 115 (17.8%) | 46 (18.9%) | 56 (20.8%) | 0.294 |
| DM with complications | 87 (13.5%) | 38 (15.6%) | 49 (18.2%) | 0.065 |
| Renal | 76 (11.8%) | 39 (16.1%) | 45 (16.7%) | 0.031 |
| Any liver | 43 (6.7%) | 12 (4.9%) | 28 (10.4%) | 0.099 |
| Severe liver | 35 (5.4%) | 11 (4.5%) | 25 (9.3%) | 0.053 |
| PUD | 20 (3.1%) | 9 (3.7%) | 20 (7.4%) | 0.005 |
| Any cancer | 58 (9.0%) | 21 (8.6%) | 12 (4.5%) | 0.026 |
| Metastatic | 16 (2.5%) | 6 (2.5%) | 1 (0.4%) | 0.055 |
| Dementia | 4 (0.6%) | 2 (0.8%) | 1 (0.4%) | 0.74 |
| CTD | 9 (1.4%) | 3 (1.2%) | 5 (1.9%) | 0.653 |
| HIV | 1 (0.2%) | 2 (0.8%) | 1 (0.4%) | 0.431 |
| Major diagnostic category groups | | | | <0.001 |
| Neurologic | | | | |
| Head and neck | 200 (31.0%) | 51 (21.0%) | 38 (14.1%) | |
| Respiratory | 7 (1.1%) | 1 (0.4%) | 3 (1.1%) | |
| Circulatory | 83 (12.9%) | 38 (15.6%) | 62 (23.1%) | |
| Gastrointestinal | 157 (24.3%) | 72 (29.6%) | 73 (27.1%) | |
| Soft Tissue | 53 (8.2%) | 18 (7.4%) | 22 (8.2%) | |
| Metabolic | 24 (3.7%) | 12 (4.9%) | 4 (1.5%) | |
| Genitourinary | 3 (0.5%) | 1 (0.4%) | 3 (1.1%) | |
| Blood/neoplastic | 7 (1.1%) | 4 (1.7%) | 3 (1.1%) | |
| Infectious Injury and intoxication | 16 (2.5%) | 3 (1.2%) | 3 (1.1%) | |
| Burns | 29 (4.5%) | 15 (6.2%) | 15 (5.6%) | |
| | 33 (5.1%) | 8 (3.3%) | 11 (4.1%) | |
| | 33 (5.1%) | 20 (8.2%) | 32 (11.9%) | |
| Treatment limitation order on admisson | 12 (1.9%) | 3 (1.2%) | 4 (1.5%) | 0.610 |
| MetroNorth Resident vs other Queensland | 284 (44.0%) | 108 (44.4%) | 119 (44.2%) | 0.939 |
| Planned admission | 127 (19.7%) | 48 (19.8%) | 58 (21.6%) | 0.802 |

*(Continued)*

**Table 1.** (Continued)

| Variable | ICU stay 14–20 days (n = 645) | ICU stay 21–27 days (n = 243) | ICU stay ≥28 days (n = 269) | p-value |
|---|---|---|---|---|
| Admission post cardiac arrest | 48 (7.5%) | 18 (7.4%) | 19 (7.1%) | 0.984 |
| Elective surgery | 80 (12.4%) | 27 (11.1%) | 26 (9.7%) | 0.504 |
| Body mass index (kg/m$^2$) | 27.8 (24.3–32.1)<br>N = 639 | 27.7 (24.1–31.9)<br>N = 240 | 28.2 (24.6–32.7)<br>N = 268 | 0.408 |
| Acute renal failure | 108 (16.7%) | 52 (21.4%) | 64 (23.8%) | 0.010 |

duration of ICU group nor the APACHE III score was associated with long-term survival. However, higher Charlson comorbidity index and increased age was associated with death.

## Discussion

In this study we report a large, multicentered study of patients requiring prolonged ICU length of stay and find that that the majority are alive with subsequent years of follow-up. Furthermore, we observed that while neither the acuity of illness at admission nor the subsequent duration of ICU stay are associated with long-term outcome, the admitting diagnosis and preceeding comorbidity are important determinants of outcome. Further studies are needed to investigate whether improved management of chronic conditions (i.e. optimization of glucose control in diabetics) prior to requirement for admission to ICU could influence their subsequent length of stay and/or outcome.

There have been several studies in the past two decades examining prolonged stays and subsequent outcome post-ICU admission [4, 22]. However, many of these studies have been limited by small sample sizes [1, 7, 10], have been single centered [2, 7, 8, 10], or have focussed on selected diagnoses or specific cohorts admitted to specialized ICUs [7]. In addition, while some national registry based studies have included very large sample sizes, they have been limited to acute care in-hospital outcomes only [3, 23]. Our study benefits from its inclusion of a large number of admissions to four ICUs representing both tertiary referral and regional hospitals. In addition, as a result of the characteristics of the included ICUs we have a broad casemix of patients represented which supports generalizability to other populations.

It is important to recognize that there is no universally accepted days of admission to define a prolonged ICU admission. We *a priori* chose to use 2 weeks as a primary definition and then to further subgroup those with 2–3 weeks and 4 or more weeks based in part on definitions used in our previous works and by others [1, 4, 11, 14, 15]. In recent years there has been increasing recognition of the concept of "persistent critical illness" for where acute transition to chronic organ failures and this is related to prolonged ICU stays. Persistent critical illness may be operationalized to occur at the point when pre-admission patients characteristics predict outcome better than acute severity of illness and diagnostic measures [3]. While the onset of persistent illness depends on the outcome measure and cohort under study, it has been estimated to onset around day 10 with a range between one and three weeks [3, 24, 25]. Although

**Table 2. Disposition of patients with prolonged ICU stay at hospital separation.**

| Discharge location | ICU stay 14–20 days (n = 527) | ICU stay 21–27 days (n = 191) | ICU stay ≥28 days (n = 224) | p-value |
|---|---|---|---|---|
| Home | 309 (58.6%) | 83 (43.5%) | 96 (42.9%) | <0.001 |
| Aged/Chronic/Palliative Care | 43 (8.2%) | 28 (14.7%) | 49 (21.9%) | <0.001 |
| Other Acute Care Hospital | 137 (26.0%) | 61 (31.9%) | 53 (23.7%) | 0.037 |
| Rehabilitation | 38 (7.2%) | 19 (10.0%) | 26 (11.6%) | 0.114 |

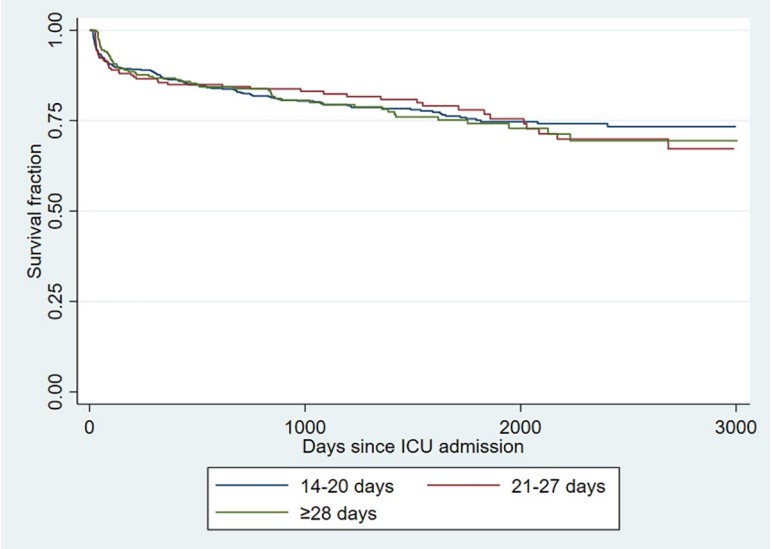

**Fig 1. Kaplan-Meier survival estimates of patients admitted to intensive care units for 14–20, 21–27, and ≥28 days.**

our study was not designed to examine this issue *per se*, the observation that co-morbidities, age, and diagnosis but not severity of illness were associated with outcome in our cohort requiring prolonged admission to ICU.

The aging of populations with increasing prevalence of chronic comorbid illnesses has and will continue to influence the contemporary epidemiology of critical illness. While there is no single "gold standard", the index developed by Charlson et al in the 1980's incorporating 17 conditions has been the most widely used to define comorbid illnesses and assess their influence on outcome related to health conditions [12]. Many revisions and modifications have been proposed to improve its discrimination and adapt to administrative data since its initial report [21, 26, 27]. In addition, others have developed schemes that incorporate a much larger number of determinants but these have had limited application in the ICU to date [26, 28]. Most commonly, studies in ICU have relied on the use of the small number of selected comorbidities included in the chronic health evaluation component of APACHE scores [6, 14]. It is important to note that while these variables have been validated for contributing to acute

**Table 3. Multivariable Cox analysis of factors associated with death among patients with prolonged admission to intensive care units.**

| Factor | Hazard ratio | 95% confidence interval | P-value |
|---|---|---|---|
| ICU stay | | | |
| 14–20 days | 1 (reference) | - | - |
| 21–27 days | 0.97 | 0.75–1.26 | 0.825 |
| ≥28 days | 0.93 | 0.72–1.20 | 0.581 |
| APACHE III score (per point) | 1.0 | 1.00–1.00 | 0.274 |
| Charlson Comorbidity Index (per point) | 1.17 | 1.13–1.22 | <0.001 |
| Age (per year) | 1.03 | 1.02–1.04 | <0.001 |
| Diagnosis | | | |
| Other | 1 (reference) | - | - |
| Gastrointestinal | 1.56 | 1.14–2.13 | 0.005 |
| Blood/neoplastic | 2.32 | 1.39–3.89 | 0.001 |

disease mortality risk, as we observed in Table 1 they may be less discriminating for examining other determinants of disease and outcome in the ICU.

Much of our focus within critical care research conducted during the past half century has been related to optimizing physiologic support and therapeutics with an emphasis on acute mortality outcomes. However, it is important to recognise that while patients with prolonged ICU stay represent a minority of all ICU admissions they consume a sizeable proportion of healthcare resources in the short and long term [3]. We observed that only one-half of patients with prolonged ICU admissions were discharged home and that more than one in five admitted to ICU for more than a month were transferred to chronic care facilities at separation from their index hospital admission. While an emphasis on mitigating the acute effects of severe critical illness will remain our priority, increasing attention to investigating the determinants and optimal management of patients requiring prolonged ICU stay is warranted.

While this study has many methodological strengths as previously noted, there are some limitations that merit discussion. First, our study was retrospective and prospective data is preferred to minimize missing data and correct application of study definitions. We did not have daily ventilator status on our entire cohort, and while we expect that nearly all patients staying in ICU for 2 weeks or more would have been ventilated we are unable to confirm this in fact. In addition, we were not able to evaluate the individual clinical decision process that was involved in relation to prolonged management of patients in the ICU. However, it is important that we utilized previously validated definitions and algorithms for establishing our other study variables [17, 18, 21]. Second, we relied on data that was available in electronic format and did not conduct individual chart reviews to confirm its validity. However, it important to note that the integrity of this data was high as fewer than 10 files were unable to be successfully linked among the nearly 30,000 admissions included. A third limitation is that we did not include complications and diagnoses (i.e. ICU acquired infections) that arose following ICU admission that could have adversely influenced patient's course and outcome. Fourth, we did not include measures of frailty. Fifth, we *a priori* chose to categorise our prolonged stays into groups rather than to analyse with days of length of stay as a continuous variable. While grouping patients makes analysis and interpretation less complicated, the possibility of loss of sensitivity to small changes in variables is raised. Sixth, although the possibility exists that race or ethnicity could have influenced our findings we did not have access to these variables. Finally, although we were able to comprehensively establish a lethal outcome in both institutional and community settings within Queensland with a high degree of certainty, patients who moved out of state and subsequently died would likely have been missed by our study methodology.

## Conclusion

This study provides novel information surrounding the determinants and outcome associated with prolonged admission to ICU in a large, mixed, Australian cohort. Further studies investigating whether optimization of chronic disease management both pre- and post ICU admission may influence the outcome of critical illness are warranted.

## Author Contributions

**Conceptualization:** Kevin B. Laupland, Mahesh Ramanan, Alexis Tabah.

**Data curation:** Mahesh Ramanan, Kiran Shekar, Felicity Edwards, Pierre Clement, Alexis Tabah.

**Formal analysis:** Kevin B. Laupland.

**Writing – original draft:** Kevin B. Laupland.

**Writing – review & editing:** Kevin B. Laupland, Mahesh Ramanan, Kiran Shekar, Felicity Edwards, Pierre Clement, Alexis Tabah.

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
