## [Decision Letter · Decision Letter 0]

18 Mar 2021

PONE-D-21-03002

Long-term outcome of prolonged critical illness: A multicentered study in North Brisbane, Australia

PLOS ONE

Dear Dr. Laupland,

Thank you for submitting your manuscript to PLOS ONE. After careful consideration, we feel that it has merit but does not fully meet PLOS ONE’s publication criteria as it currently stands. Therefore, we invite you to submit a revised version of the manuscript that addresses the points raised during the review process.

We look forward to receiving your revised manuscript.

Kind regards,

Aleksandar R. Zivkovic

Academic Editor

PLOS ONE

Journal Requirements:

4. We note you have included a table to which you do not refer in the text of your manuscript. Please ensure that you refer to Table 3 in your text; if accepted, production will need this reference to link the reader to the Table.

6. In the ethics statement in the manuscript and in the online submission form, please provide additional information about the patient records/samples used in your retrospective study.

Specifically, please ensure that you have discussed whether all data/samples were fully anonymized before you accessed them.

Reviewers' comments:

Reviewer's Responses to Questions

**Comments to the Author**

Reviewer #1: PONE-D-21-03002 review

In this study, the authors present outcomes from prolonged critical illness over multiple years and multiple hospitals in Queensland, Australia. They focused on the 1,157 patients with prolonged (>2 week) ICU admission to understand factors associated with long-term outcomes. They showed that within this group, length of prolonged stay and severity of presenting illness were not associated with long-term outcome. Chronic comorbidities and ICU presenting diagnostic categories were associated with outcome.

This study is a welcome addition to the literature and but should be strengthened by addressing the following considerations:

1. The study should include race and ethnicity breakdown of the patient population and, if racially/ethnically homogeneous population, this should be mentioned as a limitation.

2. The authors’ conclusions are compatible with previous work, cited in the Background, showing that comorbid conditions predict prolonged critical illness, which could be acknowledged in the discussion.

3. Line 64: The phrase “not limited to one or more of” is awkward and could be replaced with “including.”

4. Line 72: The term “survival experience” is inaccurate (patient experience was not assessed).

5. Line 89: limited to first ICU admission: First lifetime ICU admission? So a patient with any prior ICU admission would be excluded?

6. Line 159/Table 2: “there was significant differences (p<0.001) in the disposition by duration of ICU stay as shown in Table 2.” This is confusing as Table 2 does not include statistics, and the authors broadly conclude that duration of ICU stay was not associated with outcomes.

7. Line 172: “Table 2” should be Table 3

Reviewer #2: PONE-D-21-03002 Laupland K et al. Long-term outcome of prolonged critical illness: A multicentered study in North Brisbane, Australia

This a large retrospective study focussing on patients with an extended length of stay (LOS) in the ICU. The objective is “to describe the long-term survival and examine determinants of death among patients with prolonged ICU-admission”.

Recently, the term chronic critical illness has been used to describe patients with prolonged ICU-admissions and whose outcome is less determined by acute physiology and more by patients age and premorbid conditions. Iwashyna et al (Lancet Respir Med. 2016 Jul;4(7):566-573) noted that transition from acute to chronic critical illness occurred between day 7 and day 22 across diagnosis-based subgroups and between day 6 and day 15 across risk-of-death-based subgroups. In the present study the analysis was limited to pts with LOS >= 14 days. This seems arbitrary, more determined by the calendar than any medical or biological threshold.

Furthermore, when analysing LOS as risk factor for death, this is categorized as LOS 14-20, 21-27 and >= 28 days instead of analysing this as a continuous variable including 0-13 days, or categorised as determined by the data (e.g. tertials, quartiles).

My take on this is that the analysis would have profited by the inclusion of all patients. We know that death in the ICU is strongly affected by clinicians’ decisions to withhold or withdraw care and that a perception of an unfavourable prognosis is an important determinant of such decisions (e.g. expectation of a poor neurological outcome or unresolving organ failure). Patients who have survived to 14 days or more in the ICU have either escaped such decisions (for whatever reason) or show some sign of improvement. Thus, an arbitrary cut-off at a LOS of 14 days introduces a “survivor bias” that complicates the analysis.

Similarly, without knowledge about the decision-making process, the finding that the Charlson co-morbidity index is a risk factor for death in the study-population may reflect (possibly biased) decision making by attending physicians, rather than biological risk. In this study data on withholding care was limited to the entry of a treatment limitation order at admission and no further information about treatment limitations was available.

Reviewer #3: This study contributes to the small but growing literature on prolonged critical illness.

• Clarification of the terms persistent critical illness, prolonged critical illness, prolonged ICU stay and prolonged admission to ICU would add conceptual clarity to the research. There is substantial slippage in terminology in the literature and to some degree in the current paper and it is unclear how the authors of this paper are using these terms. I suggest the authors consider the work by Iwashyna and colleagues in providing conceptual clarity.

• I was making the assumption reading the manuscript that all patients admitted to the ICU in the study setting were also those who were mechanically ventilated and that prolonged ICU stay was the same as prolonged mechanical ventilation. However, this is not explicitly stated and clarification of whether prolonged ICU stay is synonymous/or not with prolonged mechanical ventilation would be helpful, especially for international readership. This is also important to consider when providing conceptual clarity as per my above comment.

• Several spelling and grammatical errors were noted throughout the manuscript and thus I suggest revisions to correct these errors.

Methods

• Please clarify whether all ICUs are strictly adult ICU’s or whether any include pediatric populations.

• It is unclear why inception dates differ among the various sites. Please add rationale.

• It is unclear why the authors chose the a prior specified categories of 14-20, 21-27 and greater or equal to 28 days. Please provide rational for these cut-offs and why perhaps aligning with the cut-offs recommended by others (PRoVent versus Iwashyna 2016) makes sense in the Australian context.

Results

• Please add p-values to table 2.

• P9, line 169: Please add interquartile range to (214-1888)

• P9, line 182, “improved management of chronic conditions….” Please specify when you are suggesting that improved management would occur. Prior to ICU, during ICU? What are the implication of what you are suggesting and what does this look like clinically?

6. PLOS authors have the option to publish the peer review history of their article (what does this mean?). If published, this will include your full peer review and any attached files.

Reviewer #1: **Yes: **Joanna Spencer-Segal, MD, PhD

Reviewer #2: **Yes: **Jon Henrik Laake

Reviewer #3: No

---

## [Author Response · Author response to Decision Letter 0]

22 Mar 2021

PONE-D-21-03002

Long-term outcome of prolonged critical illness: A multicentered study in North Brisbane, Australia

PLOS ONE

Dear Dr. Laupland,

Thank you for submitting your manuscript to PLOS ONE. After careful consideration, we feel that it has merit but does not fully meet PLOS ONE’s publication criteria as it currently stands. Therefore, we invite you to submit a revised version of the manuscript that addresses the points raised during the review process.

RESPONSE: Completed as instructed. 

RESPONSE: Completed as instructed. 

RESPONSE: Completed as instructed. 

We look forward to receiving your revised manuscript.

Kind regards,

Aleksandar R. Zivkovic

Academic Editor

PLOS ONE

Journal Requirements:

RESPONSE: Completed as instructed. 

RESPONSE: No references have been retracted. Reference list was checked for errors and format.

RESPONSE: We have revised the data sharing statement according to journal policy and our ethical and legal guidelines. It has been revised to “Data Availability: Data cannot be shared publicly because of institutional ethics, privacy, and confidentiality regulations. Data release for the purposes of research under Section 280 of the Public Health Act 2005 requires application to the Director General (PHA@health.qld.gov.au).” 

RESPONSE: We have revised the data sharing statement according to journal policy and our ethical and legal guidelines. It has been revised to “Data Availability: Data cannot be shared publicly because of institutional ethics, privacy, and confidentiality regulations. Data release for the purposes of research under Section 280 of the Public Health Act 2005 requires application to the Director General (PHA@health.qld.gov.au).” 

4. We note you have included a table to which you do not refer in the text of your manuscript. Please ensure that you refer to Table 3 in your text; if accepted, production will need this reference to link the reader to the Table.

RESPONSE: Reference to the table 3 in text is now made as requested. 

RESPONSE: The ORCID iD for the corresponding author has been validated. 

6. In the ethics statement in the manuscript and in the online submission form, please provide additional information about the patient records/samples used in your retrospective study.

Specifically, please ensure that you have discussed whether all data/samples were fully anonymized before you accessed them.

RESPONSE: The anonymization process for the study data has been updated in the revised ethics statement. 

Reviewers' comments:

Reviewer's Responses to Questions

Comments to the Author

Reviewer #1: PONE-D-21-03002 review

In this study, the authors present outcomes from prolonged critical illness over multiple years and multiple hospitals in Queensland, Australia. They focused on the 1,157 patients with prolonged (>2 week) ICU admission to understand factors associated with long-term outcomes. They showed that within this group, length of prolonged stay and severity of presenting illness were not associated with long-term outcome. Chronic comorbidities and ICU presenting diagnostic categories were associated with outcome.

This study is a welcome addition to the literature and but should be strengthened by addressing the following considerations:

1. The study should include race and ethnicity breakdown of the patient population and, if racially/ethnically homogeneous population, this should be mentioned as a limitation.

RESPONSE: We do not collect data on race or ethnicity. This limitation has been added to the discussion. 

2. The authors’ conclusions are compatible with previous work, cited in the Background, showing that comorbid conditions predict prolonged critical illness, which could be acknowledged in the discussion.

RESPONSE: We have further revised the discussion as recommended including an additional paragraph discussing persistent critical illness. 

3. Line 64: The phrase “not limited to one or more of” is awkward and could be replaced with “including.”

RESPONSE: Revised as recommended. 

4. Line 72: The term “survival experience” is inaccurate (patient experience was not assessed).

RESPONSE: We have changed the term to survival and removed “experience”. 

5. Line 89: limited to first ICU admission: First lifetime ICU admission? So a patient with any prior ICU admission would be excluded?

RESPONSE: We have clarified this to specify that the first admission during the inception period. 

6. Line 159/Table 2: “there was significant differences (p<0.001) in the disposition by duration of ICU stay as shown in Table 2.” This is confusing as Table 2 does not include statistics, and the authors broadly conclude that duration of ICU stay was not associated with outcomes.

RESPONSE: The p-value refers to the overall group test. This is now clarified in the text and the p-values for each discharge location have been added to the table. 

7. Line 172: “Table 2” should be Table 3

RESPONSE: We have revised the table numbering and reference to them in the text. 

Reviewer #2: PONE-D-21-03002 Laupland K et al. Long-term outcome of prolonged critical illness: A multicentered study in North Brisbane, Australia

This a large retrospective study focussing on patients with an extended length of stay (LOS) in the ICU. The objective is “to describe the long-term survival and examine determinants of death among patients with prolonged ICU-admission”.

Recently, the term chronic critical illness has been used to describe patients with prolonged ICU-admissions and whose outcome is less determined by acute physiology and more by patients age and premorbid conditions. Iwashyna et al (Lancet Respir Med. 2016 Jul;4(7):566-573) noted that transition from acute to chronic critical illness occurred between day 7 and day 22 across diagnosis-based subgroups and between day 6 and day 15 across risk-of-death-based subgroups. In the present study the analysis was limited to pts with LOS >= 14 days. This seems arbitrary, more determined by the calendar than any medical or biological threshold.

RESPONSE: We agree with the reviewer regarding the issue of timing of the transition of risk from acute to chronic (or persistent) critical illness. Studies in different jurisdictions have found variability in this timing. It was not the goal of this study to investigate the transition but rather focus on those with prolonged admissions. Two weeks admission has been used most commonly in previous studies including our own (Laupland et al Chest 2006). We chose the categories for analysis a priori based in part on this cut off but also based on other studies that have used 2, 3, and 4 weeks to define prolonged admission. We have added the rationale to the methods and discuss the issue further in the revised manuscript. 

Furthermore, when analysing LOS as risk factor for death, this is categorized as LOS 14-20, 21-27 and >= 28 days instead of analysing this as a continuous variable including 0-13 days, or categorised as determined by the data (e.g. tertials, quartiles).

RESPONSE: Patients who are admitted for short term (ie a few days, often post high risk elective surgery or overdoses or other readily reversible problems in otherwise healthy individuals) are markedly different from those that have prolonged admissions. As per our stated objectives we were not interested in examining those that did not have a prolonged admission. This is now further clarified in the revised manuscript. We have added discussion about the use of length of stay as a categorized versus continuous variable in the discussion limitations paragraph. 

My take on this is that the analysis would have profited by the inclusion of all patients. We know that death in the ICU is strongly affected by clinicians’ decisions to withhold or withdraw care and that a perception of an unfavourable prognosis is an important determinant of such decisions (e.g. expectation of a poor neurological outcome or unresolving organ failure). Patients who have survived to 14 days or more in the ICU have either escaped such decisions (for whatever reason) or show some sign of improvement. Thus, an arbitrary cut-off at a LOS of 14 days introduces a “survivor bias” that complicates the analysis.

RESPONSE: We agree with these statements. However, we did not seek to define issues surrounding a comparison of those who have shorter versus longer admissions but rather to look at the cohort with prolonged admissions. Inclusion of patients not requiring prolonged admission would be examining different questions than we sought at study design. We have added further discussion in the revised manuscript to address these concerns. 

Similarly, without knowledge about the decision-making process, the finding that the Charlson co-morbidity index is a risk factor for death in the study-population may reflect (possibly biased) decision making by attending physicians, rather than biological risk. In this study data on withholding care was limited to the entry of a treatment limitation order at admission and no further information about treatment limitations was available.

RESPONSE: We are unable to evaluate the individual decisions made by clinicians in this study. We have added discussion surrounding this in the revised manuscript. One difficulty with inclusion of treatment limitation orders that arise after admission is that they are often surrogate measures of outcome, as it is our experience that patients die late in their admission to ICU without such a treatment limitation order. Inclusion of the admission treatment goals provide a measure of the intention of the ICU management intensity. 

Reviewer #3: This study contributes to the small but growing literature on prolonged critical illness.

• Clarification of the terms persistent critical illness, prolonged critical illness, prolonged ICU stay and prolonged admission to ICU would add conceptual clarity to the research. There is substantial slippage in terminology in the literature and to some degree in the current paper and it is unclear how the authors of this paper are using these terms. I suggest the authors consider the work by Iwashyna and colleagues in providing conceptual clarity.

RESPONSE: We have revised use of the terms to be consistent throughout as recommended. We refer to prolonged ICU admission to refer to our cohort. 

• I was making the assumption reading the manuscript that all patients admitted to the ICU in the study setting were also those who were mechanically ventilated and that prolonged ICU stay was the same as prolonged mechanical ventilation. However, this is not explicitly stated and clarification of whether prolonged ICU stay is synonymous/or not with prolonged mechanical ventilation would be helpful, especially for international readership. This is also important to consider when providing conceptual clarity as per my above comment.

RESPONSE: Unfortunately, we did not have complete details surrounding ventilation status on our cohort throughout. While virtually all would have been ventilated for a prolonged period we do not have the actual data to report. We have added this limitation to the study. 

• Several spelling and grammatical errors were noted throughout the manuscript and thus I suggest revisions to correct these errors.

RESPONSE: We have rechecked the manuscript for errors and spelling throughout. 

Methods

• Please clarify whether all ICUs are strictly adult ICU’s or whether any include pediatric populations.

RESPONSE: We only included adult patients in this study. This is now emphasized in the revised text. 

• It is unclear why inception dates differ among the various sites. Please add rationale.

RESPONSE: Dates vary according to the availability of electronic data. We have added this detail to the revised manuscript. 

• It is unclear why the authors chose the a prior specified categories of 14-20, 21-27 and greater or equal to 28 days. Please provide rational for these cut-offs and why perhaps aligning with the cut-offs recommended by others (PRoVent versus Iwashyna 2016) makes sense in the Australian context.

RESPONSE: We chose these cut-off values a priori based on our past studies as well as other investigators and the practical values of 2,3, and 4 weeks. We have added discussion surrounding this as well. 

Results

• Please add p-values to table 2.

RESPONSE: Added as recommended. 

• P9, line 169: Please add interquartile range to (214-1888)

RESPONSE: Added as recommended.

• P9, line 182, “improved management of chronic conditions….” Please specify when you are suggesting that improved management would occur. Prior to ICU, during ICU? What are the implication of what you are suggesting and what does this look like clinically?

RESPONSE: This is only speculative. We have further clarified this in the revised manuscript to indicate this and suggest its further exploration. 

6. PLOS authors have the option to publish the peer review history of their article (what does this mean?). If published, this will include your full peer review and any attached files.

Do you want your identity to be public for this peer review? For information about this choice, including consent withdrawal, please see our Privacy Policy.

Reviewer #1: Yes: Joanna Spencer-Segal, MD, PhD

Reviewer #2: Yes: Jon Henrik Laake

Reviewer #3: No

RESPONSE: We have used the PACE application as requested.

---

## [Editor Report · Decision Letter 1]

26 Mar 2021

Long-term outcome of prolonged critical illness: A multicentered study in North Brisbane, Australia

PONE-D-21-03002R1

Dear Dr. Laupland,

We’re pleased to inform you that your manuscript has been judged scientifically suitable for publication and will be formally accepted for publication once it meets all outstanding technical requirements.

Kind regards,

Aleksandar R. Zivkovic

Academic Editor

PLOS ONE

---

## [Editor Report · Acceptance letter]

30 Mar 2021

PONE-D-21-03002R1 

Long-term outcome of prolonged critical illness: A multicentered study in North Brisbane, Australia 

Dear Dr. Laupland:

I'm pleased to inform you that your manuscript has been deemed suitable for publication in PLOS ONE. Congratulations! Your manuscript is now with our production department. 

Kind regards, 

on behalf of

Dr. Aleksandar R. Zivkovic 

Academic Editor

PLOS ONE